# Respecting Transfer Gap in Knowledge Distillation

**Yulei Niu**[*1]    **Long Chen**[1]    **Chang Zhou**[2]    **Hanwang Zhang**[3]
[1]Columbia University   [2]Damo Academy, Alibaba Group   [3]Nanyang Technological University
{yn.yuleiniu,zjuchenlong}@gmail.com   zhouchang.zc@alibaba-inc.com
hanwangzhang@ntu.edu.sg

## Abstract

Knowledge distillation (KD) is essentially a process of transferring a teacher model's behavior, *e.g.*, network response, to a student model. The network response serves as additional supervision to formulate the machine domain (*machine* for short), which uses the data collected from the human domain (*human* for short) as a transfer set. Traditional KD methods hold an underlying assumption that the data collected in both human domain and machine domain are both independent and identically distributed (IID). We point out that this naïve assumption is unrealistic and there is indeed a transfer gap between the two domains. Although the gap offers the student model external knowledge from the machine domain, the imbalanced teacher knowledge would make us incorrectly estimate how much to transfer from teacher to student per sample on the non-IID transfer set. To tackle this challenge, we propose Inverse Probability Weighting Distillation (IPWD) that estimates the propensity score of a training sample belonging to the machine domain, and assigns its inverse amount to compensate for under-represented samples. Experiments on CIFAR-100 and ImageNet demonstrate the effectiveness of IPWD for both two-stage distillation and one-stage self-distillation.

## 1   Introduction

Knowledge distillation (KD) [20] transfers knowledge from a teacher model, *e.g.*, a big, cumbersome, and energy-inefficient network, to a student model, *e.g.*, a small, light, and energy-efficient network, to improve the performance of the student model. A common intuition is that a teacher with better performance will teach a stronger student. However, recent studies find that the teacher's accuracy is not a good indicator of the resultant student performance [8]. For example, a poorly-trained teacher with early stopping can still teach a better student [8, 11, 67]; or, a teacher with a smaller model size than the student is also a good teacher [67]; or, a teacher with the same architecture as the student helps to improve the student—self-distillation [13, 71, 70, 25].

Should we view KD in the perspective of domain transfer [12, 55], we would better understand the above counter-intuitive findings. From Figure 1, we can see that teacher predictions and ground-truth labels indeed behave differently. Although the teacher is trained on the balanced dataset, its predicted probability distribution over the dataset is imbalanced. Even on the same training set with the same model parameter, teachers with different temperature $\tau$ yield different "soft label" distributions from the ground-truth ones. This implies that human and teacher knowledge is from different domains, and there is a transfer gap that drives the "dark knowledge" [20] transferring from teacher to student—regardless of "strong" or "weak" teachers, it is a valid transfer as long as there is a gap. However, the transfer gap affects the distillation performance of the under-represented classes, *i.e.*, classes on the tail of teacher predictions, which is overlooked in recent studies. Take CIFAR-100 as an example. We rank and divide the 100 classes into four groups according to the ranks of predicted

---

*Work done when Yulei was at Nanyang Technological University.

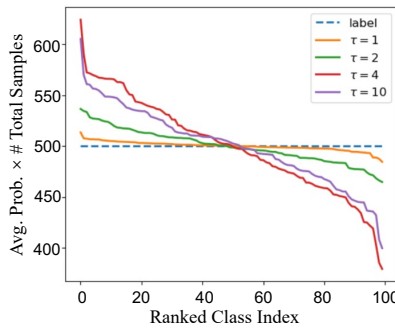 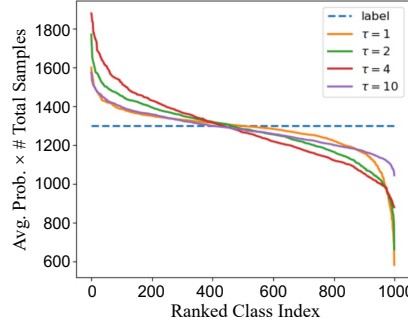

Dataset: CIFAR-100; Teacher: ResNet-110          Dataset: ImageNet; Teacher: ResNet-50

Figure 1: Illustration of the distribution discrepancies among ground-truth annotations and teacher predictions. Although the teacher model is trained on the balanced data (blue dashed), its prediction distributions are imbalanced according to various temperatures.

probability. As shown in Table 1, compared to vanilla training, KD achieves better performance in all the subgroups. However, the increase in the top 25 classes is much higher than that in the last 25 classes, *i.e.*, averagely 5.14% vs. 0.85%. We ask: what causes the gap in the first place; or more specifically, why does the teacher's non-uniform distributed predictions implies the gap? We answer from an *invariance vs. equivariance* learning point of view [4, 60]:

**Human domain: context invariance**. The discriminative generalization is the ability to learn both context-invariant and class-equivariant information from the diverse training samples per class. The human domain only provides context-invariant class-specific information, *i.e.*, hard targets. We normally collect a balanced dataset to formulate human domain.

Table 1: Improvement of KD over vanilla student for different classes. The metric is macro-average recall.

| Arch. style | Top 1-25 | Top 26-50 | Top 51-75 | Top 76-100 |
|---|---|---|---|---|
| ResNet50 -> MobileNetV2 | +4.96 | +5.92 | +1.76 | +1.20 |
| resnet32x4 -> ShuffleNetV1 | +5.80 | +2.68 | +2.52 | +0.84 |
| resnet32x4 -> ShuffleNetV2 | +4.72 | +1.92 | +2.24 | +0.76 |
| WRN-40-2 -> ShuffleNetV1 | +5.08 | +7.20 | +4.48 | +0.60 |

**Machine domain: context equivariance**. Teacher models often use a temperature variable to preserve the context. The temperature allows the teacher to represent a sample not only by its context-invariant class-specific information, but also its context-equivariant information. For example, a dog image with a soft label 0.8·dog + 0.2·wolf may imply that the dog has wolf-like contextual attributes such as "fluffy coat" and "upright ears". Although the context-invariance (*i.e.*, class) is balanced in the training data, the context-equivariance (*i.e.*, context) is imbalanced because the context balance is not considered in class-specific data collection [58]. To construct the transfer set for the machine domain, the teacher model annotates each sample after seeing others, *i.e.*, being pre-trained on the whole set. Interestingly, the diverse context results in a long-tailed imbalanced distribution, which is exactly reflected in Figure 1. In other words, the teacher's knowledge is imbalanced even though the teacher is trained on a class-balanced dataset.

Now we are ready to point out how the transfer gap is not properly addressed in conventional KD methods. Conventional KD calculates the Cross-Entropy (CE) loss between the ground-truth label and the student's prediction, and the Kullback–Leibler (KL) divergence [30] loss between the teacher's and student's predictions, where a constant weight is assigned for the two losses. This is essentially based on the underlying assumption that the data in both the human and machine domains are IID. Based on the analysis of context equivariance, we argue that the assumption is unrealistic, *i.e.*, *the teacher's knowledge is imbalanced*. Therefore, a constant sample weight for the KL loss would be a bottleneck. In this paper, we propose a simple yet effective method, Inverse Probability Weighting Distillation (IPWD), which compensates for the training samples that are under-weighted in the machine domain. For each training sample $x$, we first estimate its machine-domain propensity score $P(x|machine)$ by comparing class-aware and context-aware predictions. A sample with a low propensity score would have a high confidence from class-aware predictions and a low confidence from context-aware predictions. Then, IPWD assigns the inverse probability $1/P(x|machine)$ as the sample weight for the KL loss to highlight the under-represented samples. In this way, IPWD generates a pseudo-population [34, 24] to deal with imbalanced knowledge.

We evaluate our proposed IPWD on two typical knowledge distillation settings: two-stage teacher-student distillation and one-stage self-distillation. Experiments conducted on CIFAR-100 [29] and ImageNet [10] demonstrate the effectiveness and generality of our IPWD.

Our contributions are three-fold:

- We formulate KD as a domain transfer problem and argue that the naïve IID assumption on machine domain neglects the imbalanced knowledge due to the transfer gap.

- We propose Inverse Probability Weighting Distillation (IPWD) which compensate for the samples that are overlooked in the machine domain to tackle the imbalanced knowledge in transfer gap.

- Experiments on CIFAR-100 and ImageNet for both two-stage distillation and one-stage self-distillation show that the proper handling of the transfer gap is a promising direction in KD.

## 2 Related Work

**Knowledge distillation** (KD) was first introduced to transfer knowledge from an effective but cumbersome model to a smaller and more efficient model [20]. The knowledge can be formulated in either output space [20, 26, 32, 68, 67, 38, 53, 73, 28] or representation space [48, 23, 69, 27, 44, 18, 57, 7, 25]. KD has attracted a wide interest in theory, methodology, and applications [14]. For applications, KD has shown its great potential in various areas, including but not limited to classification [33, 47, 36, 21], detection [31, 52, 61], segmentation [17, 39, 35] for visual recognition tasks, and visual question answering [41, 1, 42], video captioning [43, 72], and text-to-image synthesis [56] for vision-language tasks. Recent studies further discussed how and why KD works. Specifically, Müller *et al.* [40] and Shen *et al.* [51] empirically analyzed the effect of label smoothing on KD. Cho *et al.* [8], Dong *et al.* [11], and Yuan *et al.* [67] pointed out that early stopping is a good regularization for a better teacher. Yuan *et al.* [67] further found that a poorly trained teacher, even a model smaller than the student, can improve the performance of the student. Besides, Memon *et al.* [37] and Zhou *et al.* [73] proposed a bias-variance trade-off perspective for KD. In this paper, we point out that existing KD methods hold an underlying assumption that the IID training samples are also IID in the machine domain, which overlooks the transfer gap.

**Self-distillation** is a special case of KD, which uses the student network itself as the teacher instead of the cumbersome model, *i.e.*, the teacher and student models have the same architecture [13, 71, 70, 25]. This process can be executed in iterations and produce a stronger ensemble model [13]. Similar to KD, traditional self-distillation follows a two-stage process: first pre-training a student model as the teacher, and then distilling the knowledge from the pre-trained model to a new student model. In order to perform the teacher-student optimization in one generation, recent studies [65, 28] proposed one-stage self-distillation that adopts student models at earlier epochs as teacher models. These one-stage self-distillation methods outperform vanilla students by large margins. In this paper, we also evaluate the effectiveness of our IPWD as a plug-in in one-stage self-distillation.

**Inverse Probability Weighting** (IPW) [49, 34, 24, 5], also known as inverse probability of treatment weighting or inverse propensity weighting, was proposed to correct the selection bias when the observations are non-IID. IPW uses the inverse of the probability (*i.e.*, propensity score) that the individual would be assigned to the treatment group to reweight the samples. Propensity-weighting techniques have been widely applied and studied in many areas [50], such as causal inference [24], complete-case analysis [34], machine learning [9, 6, 54], and recommendation systems [50, 62, 3]. In this paper, we view the distillation process as a domain transfer problem and adopt IPW to dynamically assign the weight to each training sample for the distillation loss.

## 3 Analysis

### 3.1 Knowledge Distillation (KD)

We view knowledge distillation from a perspective of domain transfer, and take the image classification task as the case study. Suppose that the training data $\mathcal{D} = \{\mathcal{X}, \mathcal{Y}\} = \{(x, y)\}$ contains $x$ as the input (*e.g.*, image) and $y \in \mathbb{R}^C$ as its ground-truth annotation (*e.g.*, one-hot label), where $C$ denotes the number of classes. A standard solution to train the classifier $\theta$ uses the cross-entropy loss as the

objective:

$$\mathcal{L}_{cls}(human;\theta) = \mathbb{E}_{(x,y)\sim P_{human}}[\ell_{cls}(x,y;\theta)] \approx \frac{1}{|\mathcal{D}|}\sum_{(x,y)\in\mathcal{D}}\ell_{cls}(x,y;\theta) \triangleq \mathcal{L}_{cls}(\mathcal{D};\theta), \quad (1)$$

where $\ell_{cls}(x,y) = H(y^s,y)$ is the classification loss for sample $x$, $H(p,q) = \sum_{i=1}^{C} -q_i\log p_i$ denotes the cross entropy between $p$ and $q$, $y^s = f(x;\theta)$ denotes the model's output probability given $x$, i.e., $y_k^s = \frac{\exp(z_k^s)}{\sum_{i=1}^{C}\exp(z_i^s)}$, where $z^s$ is the output logits of the model. The hard targets provide context-invariant class-specific information from the human domain. An assumption held behind Eq. (1) is that the samples are independent and identically distributed (IID) in the training and test set.

KD adopts a teacher model $\theta^t$ to generate soft targets as extra supervisions, i.e., context-equivariant information. To formulate the machine domain, traditional KD methods commonly use the training set $\mathcal{D}$ to construct the transfer set $\mathcal{D}^t$ using the same copy of $\mathcal{X}$, i.e., $\mathcal{D}^t = \{(x,y^t)\}$ where $y^t = f(x;\theta^t)$ and $x\in\mathcal{X}$. Traditional KD approaches use the KL divergence [30] loss for knowledge transfer:

$$\mathcal{L}_{dist}(machine;\theta) = \mathbb{E}_{(x,y)\sim P_{machine}}[\ell_{dist}(x,y;\theta)] \approx \frac{1}{|\mathcal{D}^t|}\sum_{(x,y^t)\in\mathcal{D}^t}\ell_{dist}(x,y^t;\theta) \triangleq \mathcal{L}_{dist}(\mathcal{D}^t;\theta),$$
$$(2)$$

where $\ell_{dist}(x,y^t;\theta) = \tau^2\cdot[H(y_\tau^s,y_\tau^t) - H(y_\tau^t,y_\tau^t)]$ denotes the distillation loss for sample $x$. Normally, the outputs of the student and teacher are softened using a temperature $\tau$, i.e., $y_{\tau,k}^s = \frac{\exp(z_k^s/\tau)}{\sum_{i=1}^{C}\exp(z_i^s/\tau)}$ and $y_{\tau,k}^t = \frac{\exp(z_k^t/\tau)}{\sum_{i=1}^{C}\exp(z_i^t/\tau)}$. The overall objective combines $\mathcal{L}_{cls}$ and $\mathcal{L}_{dist}$ as:

$$\mathcal{L}_{kd} = \alpha\cdot\mathcal{L}_{cls} + \beta\cdot\mathcal{L}_{dist}, \quad (3)$$

where $\alpha$ and $\beta$ are the hyper-parameters. The underlying assumption of traditional KD behind Eq. (2) is that the transfer set $\mathcal{D}^t$ is an unbiased approximation of the machine domain. However, the observed long-tailed and temperature-sensitive distributions of teacher's predictions in Figure 1 rationally challenge this assumption. As a result, samples with lower $P(x|machine)$ are under-represented during the distillation process, which affects the unbiasedness of knowledge transfer. This analysis indicates that Eq. (2) is not optimal to utilize the teacher's imbalanced knowledge.

### 3.2 Transfer Gap in KD

We interpret the transfer gap and its confounding effect from the perspective of causal inference. Figure 2 illustrates the causal relations between the image $X$, training data $\mathcal{D} = \{(x,y)\}$, teacher's parameters $\theta^t$ and teacher's output $Y^t$ in KD. Overall, $\mathcal{D}$ and $\theta^t$ jointly act as the confounder of $X$ and $Y^t$ in the transfer set. First, the training set $\mathcal{D}$ and transfer set of teacher

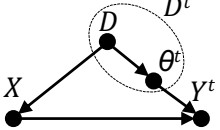

Figure 2: Causal graph for KD.

model $\mathcal{D}^t = \{(x,y^t)\}$ share the same image set, and $X = x$ is sampled from the image set of $\mathcal{D}$, i.e., $\mathcal{D}$ serves the cause of $X$. Second, the teacher $\theta^t$ is trained on $\mathcal{D}$, and $y^t$ is calculated based on $\theta^t$ and $x$, i.e., $y^t = f(x;\theta^t)$. Therefore, $X$ and $\theta^t$ are the cause of $Y^t$. Note that the transfer set is constructed based on the images on $\mathcal{D}$ and teacher model $\theta^t$. Therefore, we regard the transfer set $\mathcal{D}^t$, the joint of $\mathcal{D}$ and $\theta^t$, as the confounder of $X$ and $Y^t$.

Although $\mathcal{D}$ is balanced when considering the context-invariant class-specific information, the context information (e.g., attributes) is overlooked, which makes the $\mathcal{D}$ imbalanced on context. As shown in Figure 1, such an imbalanced context leads to an imbalanced transfer set $\mathcal{D}^t$ and further affects the distillation performance of teacher's knowledge.

To overcome such confounding effect, a commonly used technique is intervention via $P(y^t|do(x))$ instead of $P(y^t|x)$, which is formulated as $P(y^t|do(x)) = \sum_{\mathcal{D}^t} P(y|x,\mathcal{D}^t)P(\mathcal{D}^t) = \sum_{\mathcal{D}^t}\frac{P(x,y^t,\mathcal{D}^t)}{P(x|\mathcal{D}^t)}$. This transformation suggests that we can use the inverse of propensity score, $1/P(x|\mathcal{D}^t)$, as sample weight to implement the intervention and overcome the confounding effect. Thanks to the causality-based theory [49, 5], we can use the Inverse Probability Weighting (IPW) technique to overcome the confounding effect brought by the transfer gap.

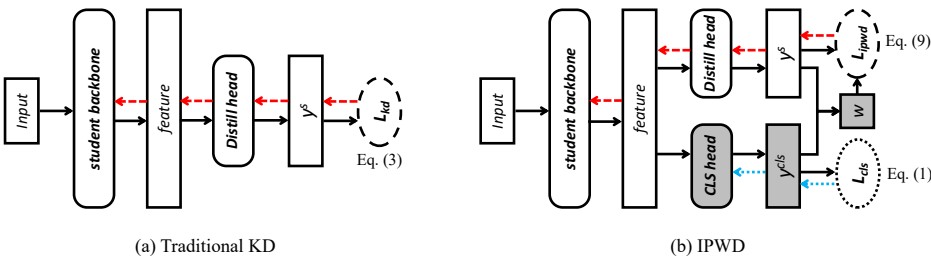

(a) Traditional KD  (b) IPWD

Figure 3: The comparison of training pipelines between traditional KD and our IPWD. The module and outputs in grey are not used in the delivered student model.

# 4 Method

We propose a simple yet effective method, Inverse Probability Weighting Distillation (IPWD), to respect the transfer gap and imbalance knowledge in KD. In this section, we first introduce the overall framework of IPWD, then present the implementation details.

## 4.1 Inverse Probability Weighting for KD

As analyzed in Section 3, the IID training samples in the human domain are no longer IID in the machine domain. Simply assuming the training set as the perfect transfer set may lead to the selection bias: samples that match "head" knowledge are over-represented and easy to be observed, while samples that match "tail" knowledge are under-represented and hard to be observed. This would suppress the transfer of "tail" knowledge. The analysis from the perspective of causal inference in Section 3.2 suggests that we can use Inverse Probability Weighting (IPW) for debiased distillation. In short, IPW generates a pseudo-population where under-represented samples are assigned with large weights and over-represented samples are assigned with small weights. The weight for sample $x$ is determined as the inverse of its probability, also known as propensity score, to the domain $d \in \{human, machine\}$, $i.e.$, $w_{x|d} = 1/p(x|d)$. We adopt IPW to KD and obtain the following objective for sample $x$:

$$\ell(x;\theta) = \sum_d w_{x|d} \cdot \ell_d(x, y^d; \theta) = \frac{1}{P(x|human)} \ell_{cls}(x, y; \theta) + \frac{1}{P(x|machine)} \ell_{dist}(x, y^t; \theta). \quad (4)$$

## 4.2 Implementation

Since the training and test data are normally IID in the human domain, we safely and rationally use the empirical risk. Therefore, we assign a constant weight to each sample when calculating the classification loss. For sure, the training and test samples can be both non-IID, $e.g.$, long-tailed recognition tasks, which is out of the scope of this paper.

As analyzed in Section 1, the assumption held by traditional KD, $i.e.$, both $P(X|human)$ and $P(X|machine)$ are IID, is unrealistic in practice. Therefore, we should consider the propensity score $P(x|machine)$ as a sample-specific value for the distillation loss to improve the generalization. Traditional IPW estimates the propensity score using logistic regression, $i.e.$, $\hat{P}(x|machine) = 1/(1 + \exp(-z_x))$, where $z_x$ is the logit for $x$. Since the ground-truth annotation of $P(x|machine)$ is not available, it is not practicable to directly train the regression model in a fully-supervised manner. Therefore, we estimate the propensity score in an unsupervised way.

Recall that the samples with high propensity are over-represented in the transfer set. As a result, the student model would learn less from the under-represented group via distillation. Therefore, we use a classification-trained (CLS-trained) classifier for the human domain as reference, and assume that a KD-trained classifier for the machine domain is more confident for the over-represented group than the CLS-trained classifier. We compare the outputs of two classifiers to identify whether a sample is under-represented in the machine domain. Suppose that the KD-trained output is $y^{kd}$ and the CLS-trained output is $y^{cls}$. The assumption implies that the logit $z_x$ is negatively correlated with

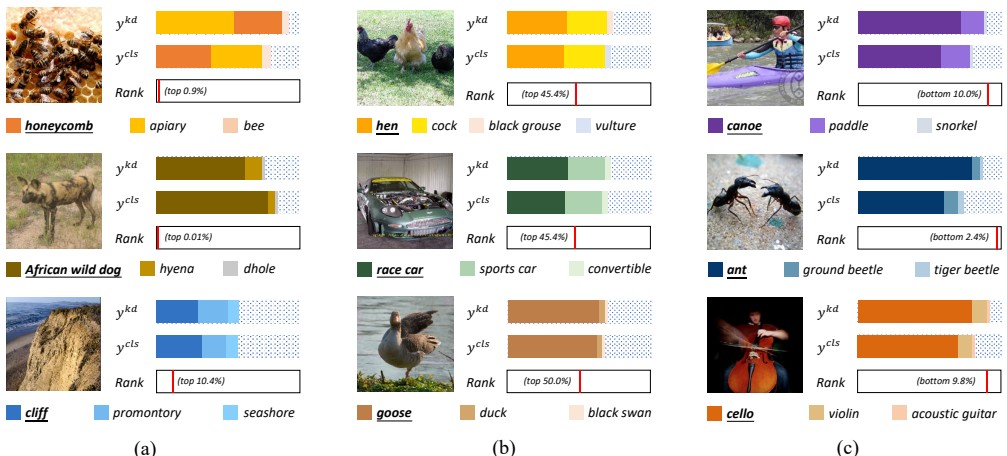

Figure 4: Illustration of the sample weights. **Bold underlined class** denotes the ground-truth label. The areas in $y^{kd}$ and $y^{cls}$ represent the proportion of the predicted probability. A high rank indicates a large sample weight, while a low rank indicates a small sample weight.

$H(y^{kd}, y)$ and positively correlated with $H(y^{cls}, y)$, where $H(\cdot, \cdot)$ is the cross-entropy defined in Section 3. Considering the range of logit, we estimate $z_x$ as $z_x = \log \frac{H(y^{cls}, y)}{H(y^{kd}, y)}$.

Figure 3 illustrates the comparison between traditional KD and our IPWD. We take the KD-trained student's output $y^s$ as $y^{kd}$, and train an extra classifier head (*i.e.*, "CLS head" in Figure 3(b)) to calculate $y^{cls}$, which is optimized using the cross-entropy loss $\mathcal{L}_{cls}$ with the ground-truth labels. As shown in Table 5, we empirically found that directly using $y^s$ and $y^{cls}$ would lead to a high variance. For example, a wrongly classified sample may have an extremely large loss and heavily suppress the distillation of other samples. Therefore, we normalize the logits by dividing them by the standard deviation $\sigma^s$ and $\sigma^{cls}$, *i.e.*, $\widetilde{y}^s = \frac{\exp(z_k^s/\sigma^s)}{\sum_{i=1}^{C} \exp(z_i^s/\sigma^s)}$ and $\widetilde{y}^{cls} = \frac{\exp(z_k^{cls}/\sigma^{cls})}{\sum_{i=1}^{C} \exp(z_i^{cls}/\sigma^{cls})}$. In this way, the outputs are at the same scale with a standard deviation equal to 1, which helps to reduce the variance. We finally take $\widetilde{y}^s$ as $y^{kd}$ and $\widetilde{y}^{cls}$ as $y^{cls}$. Combining $z_x$ into the propensity score, we have:

$$\hat{P}(x|machine) = \frac{H(y^{cls}, y)}{H(y^{cls}, y) + H(y^{kd}, y)}, \qquad (5)$$

and the estimated weight $\hat{w}_x$ for sample $x$ is $\hat{w}_x = \frac{1}{\hat{P}(x|machine)} = 1 + \frac{H(y^{kd}, y)}{H(y^{cls}, y)}$. Figure 4 illustrates some examples of training samples and their assigned weights. Under-represented samples, for which the KD-trained classifier is less confident than the CLS-trained classifier, are assigned with a large weight (Figure 4(a)). Over-represented samples, for which the KD-trained classifier is more confident than the CLS-trained classifier, are assigned with a small weight (Figure 4(c)). Samples for which the two classifiers behave similarly are assigned with a balanced weight (Figure 4(b)). The weighted distillation loss is formulated as:

$$\mathcal{L}_{ipw\text{-}dist} = \frac{1}{|\mathcal{D}^t|} \sum_{(x, y^t) \in \mathcal{D}^t} \hat{w}_x \cdot \ell_{dist}(x, y^t; \theta), \qquad (6)$$

Our final IPWD objective is formulated as:

$$\mathcal{L}_{ipwd} = \mathcal{L}_{cls} + \alpha \mathcal{L}_{ipw\text{-}dist} \qquad (7)$$

where $\alpha$ is a trade-off hyper-parameter between classification and distillation.

**Limitations and negative societal impacts.** As introduced in Section 4.2, we estimate the propensity score by comparing the heads of the student model. Therefore, the estimation relies on the quality of the student model. A poor student may not correctly estimate the propensity, which may further suppress the effectiveness of IPWD. Also, we assume that the training and test samples are IID in the human domain, which may not be valid for long-tailed tasks. To the best of our knowledge, as our work is purely an algorithm for knowledge distillation, we haven't found any negative societal impact.

Table 2: Top-1 accuracies (%) on CIFAR-100 for two-stage distillation. * denotes our reproduced results using the same teacher model.

| Teacher
Student | Same architecture style | | | | Different architecture style | | | |
|---|---|---|---|---|---|---|---|---|
| | WRN-40-2
WRN-40-1 | resnet56
resnet20 | resnet110
resnet32 | resnet32x4
resnet8x4 | resnet32x4
ShuffleNetV1 | WRN-40-2
ShuffleNetV1 | ResNet50
vgg8 | ResNet50
MobileNetV2 |
| Teacher | 75.61 | 72.34 | 74.31 | 79.42 | 79.42 | 75.61 | 79.34 | 79.34 |
| Student | 71.98 | 69.06 | 71.14 | 72.50 | 70.50 | 70.50 | 70.36 | 64.60 |
| FitNet [48] | 72.24 | 69.06 | 71.06 | 73.50 | 73.59 | 73.73 | 70.69 | 63.16 |
| AT [69] | 72.77 | 69.21 | 72.31 | 73.44 | 71.73 | 73.32 | 71.84 | 58.58 |
| SP [59] | 72.43 | 69.67 | 72.69 | 72.94 | 73.48 | 74.52 | 73.34 | 68.08 |
| CC [46] | 72.21 | 69.63 | 71.48 | 72.97 | 71.14 | 71.38 | 70.25 | 65.43 |
| VID [2] | 73.30 | 70.38 | 72.61 | 73.09 | 73.38 | 73.61 | 70.30 | 67.57 |
| RKD [44] | 72.22 | 69.61 | 71.82 | 71.90 | 72.28 | 72.21 | 71.50 | 64.43 |
| PKT [45] | 73.45 | 70.34 | 72.61 | 73.64 | 74.10 | 73.89 | 73.01 | 66.52 |
| AB [19] | 72.38 | 69.47 | 70.98 | 73.17 | 73.55 | 73.34 | 70.65 | 67.20 |
| FT [27] | 71.59 | 69.84 | 72.37 | 72.86 | 71.75 | 72.03 | 70.29 | 60.99 |
| NST [23] | 72.24 | 69.60 | 71.96 | 73.30 | 74.12 | 74.89 | 71.28 | 64.96 |
| KD [20] | 73.54 | 70.66 | 73.08 | 73.33 | 74.07 | 74.83 | 73.81 | 67.35 |
| CRD [57] | 74.14 | 71.16 | 73.48 | 75.51 | 75.11 | 76.05 | 74.30 | 69.11 |
| WSLD* [73] | 73.74 | **71.53** | 73.36 | 74.79 | 75.09 | 75.23 | 73.80 | 68.79 |
| **IPWD** | **74.64** | 71.32 | **73.91** | **76.03** | **76.03** | **76.44** | **74.97** | **70.25** |

## 5 Experiments

We take the image classification task as a case study to evaluate the effectiveness and generalizability of our IPWD. Following previous works [57, 73, 28], we conduct experiments with two settings, two-stage distillation and one-stage self-distillation.

### 5.1 Datasets and Settings

**Datasets**. We conducted experiments on CIFAR-100 [29] and ImageNet [10]. CIFAR-100 contains 50K images in the training set and 10K images in the test set from 100 classes. ImageNet provides 1.2M images in the training set and 50K images in the validation set from 1K classes.

**Settings**. Two-stage distillation is the conventional setting that pre-trains a teacher model at the first stage and transfers the knowledge to a student model at the second stage. Commonly, the teacher is a larger model, and the student is a smaller model. For self-distillation, the teacher and student have the same architecture. One-stage self-distillation aims to complete the teacher-student optimization simultaneously [65, 28], *i.e.*, the pre-training and transfer processes are reduced to one.

### 5.2 Two-stage Distillation

**Baseline methods**. For two-stage distillation, following Tian *et al.* [57] and Zhou *et al.* [73], we considered the following methods as baselines: KD [20], FitNet [48], AT [69], SP [59], CC [46], VID [2], RKD [44], PKT [45], FSP [66], AB [19], FT [27], NST [23], CRD [57], SSKD [64], and WSLD [73]. In particular, WSLD [73] is the most related work to us, which proposed a bias-variance trade-off perspective for KD and also assigns different weights to each training sample. Similarly, the weight is positive related to the cross-entropy loss of student's output. The main differences between our IPWD and WSLD are as follows. First, our formulation of samples weights is theoretically guaranteed by the causal theory behind Inverse Probability Weighting (IPW) [49, 34, 24, 5]. Second, WSLD estimates the sample weight using both the student model and the teacher model. As a comparison, we use the student model with two different classifier heads to guarantee that the capacities of the compared models are close.

**Implementation**. For experiments on CIFAR-100, we followed CRD [57] based on the open-sourced code. We set the trade-off hyper-parameter $\alpha = 5$ in Eq. (7) and the temperature $\tau = 10$. Other training details were the same as CRD [57] and provided in the appendix. For ImageNet, we followed

Table 3: Top-1 accuracies (%) on CIFAR-100 test set as a plug-in on SSKD [64]. We reproduced the results of SSKD using the same teacher model.

| Teacher Student | Same architecture style | | | | Different architecture style | | | |
|---|---|---|---|---|---|---|---|---|
| | WRN-40-2 WRN-16-2 | WRN-40-2 WRN-40-1 | resnet56 resnet20 | resnet32x4 resnet8x4 | ResNet50 MobileNetV2 | resnet32x4 ShuffleNetV1 | WRN-40-2 ShuffleNetV1 | vgg13 MobileNetV2 |
| Teacher | 76.46 | 76.46 | 73.44 | 79.63 | 79.10 | 79.63 | 76.46 | 75.38 |
| Student | 73.64 | 72.24 | 69.63 | 72.51 | 65.79 | 70.77 | 70.77 | 65.79 |
| SSKD* [64] | 75.74 | 75.59 | 70.61 | 75.80 | 72.22 | 77.71 | 78.49 | 77.32 |
| **+ IPWD** | **76.39** | **76.09** | **71.69** | **76.74** | **72.85** | **78.30** | **79.17** | **77.95** |

Zhou *et al.* [73] to conduct experiments based on their open-sourced code. We used the same hyper parameters as WSLD [73], *i.e.*, $\alpha$ as 2.5 and $\tau$ as 2.

**Comparison with baseline methods**. Table 2 shows the results of student models on CIFAR-100 with different teacher-student architectures, which can be grouped into same architecture style and different architecture style. Note that the results of WSLD reported in [73] used a different pre-trained teacher model. Since some training techniques like early-stopping [8, 11, 67] may improve the distillation performance, we reimplemented WSLD using the same teacher model for a fair comparison. Overall, our IPWD outperforms KD by large margins and outperforms other baseline methods on most of the architectures, which demonstrates the effectiveness of our IPWD. In particular, the improvement with the same architecture style is smaller than the different style. The reason is that the different architecture style reflects the bigger gap between the human domain and machine domain. Since our IPWD weights the training samples to address the non-IID problem, IPWD successfully outperforms KD and other state-of-the-art methods by large margins when the transfer gap is significant.

Note that SSKD [64] achieves higher performance because of (1) a better teacher model, and (2) data augmentation for structured knowledge distillation. We further apply IPWD to SSKD as a plug-in by weighting the logit distillation objective and keeping the structured knowledge distillation terms unchanged. Table 3 shows that our IPWD can consistently improve SSKD by 0.5~1.0% for different architectures. These results indicate that our IPWD is a good complementary to distillation methods.

Table 4 further shows the comparison on ImageNet. Following CRD [57] and WSLD [73], we used two teacher-student architectures as the representatives. For the same architecture style, our IPWD improves KD by 1.21%, and achieves competitive performance compared to WSLD. For the different architecture style, the improvement of WSLD over KD drops from 1.37% to 1.03%. As a comparison, our IPWD improves KD

Table 4: Acc. (%) on ImageNet for two-stage distillation.

| Teacher Student | Same arch. style | | Diff. arch. style | |
|---|---|---|---|---|
| | ResNet-34 ResNet-18 | | ResNet-50 MobileNet-v1 | |
| | **Top-1** | **Top-5** | **Top-1** | **Top-5** |
| Teacher | 73.31 | 91.42 | 76.16 | 92.87 |
| Student | 69.75 | 89.07 | 68.87 | 88.76 |
| AT [69] | 71.03 | 90.04 | 70.18 | 89.68 |
| NST [23] | 70.29 | 89.53 | — | — |
| FT [27] | — | — | 69.88 | 89.50 |
| FSP [66] | 70.58 | 89.61 | — | — |
| AB [19] | — | — | 68.89 | 88.71 |
| RKD [44] | 70.40 | 89.78 | 68.50 | 88.32 |
| KD [20] | 70.67 | 90.04 | 70.49 | 89.92 |
| Overhaul [18] | 71.03 | 90.15 | 71.33 | 90.33 |
| CRD [57] | 71.17 | 90.13 | 69.07 | 88.94 |
| SSKD [64] | 71.62 | 90.67 | — | — |
| DGKD [53] | 71.73 | **90.82** | — | — |
| WSLD [73] | **72.04** | 90.70 | 71.52 | 90.34 |
| **IPWD** | 71.88 | 90.50 | **72.65** | **91.08** |

by 2.16%, and outperforms WSLD by 1.13%. This improvement on the large-scale dataset further demonstrates the effectiveness of our IPWD in bridging the transfer gap when the student and teacher model have different architecture styles, which is more practical in real-world applications.

**Ablation study: technical designs**. As introduced in Section 4.2, we used an extra classifier head to produce CLS-trained output, and normalized the logits to reduce the variance for propensity estimation. Note that WSLD [73] uses the teacher model to estimate the sample weight, and the teacher model is also trained with the cross-entropy loss. Therefore, we considered an alternative which replaces the classification head with the teacher model to produce the classification-aware

Table 5: Ablation study of technical designs for weight estimation on CIFAR-100. "CLS head" denotes the usage of an extra classification head. "logits norm." denotes that the logits are normalized before calculating the propensity.

| | CLS head | logits norm. | Same architecture style | | | Different architecture style | | |
|---|---|---|---|---|---|---|---|---|
| | | | WRN-40-2 ↓ WRN-40-1 | resnet110 ↓ resnet32 | resnet32x4 ↓ resnet8x4 | resnet32x4 ↓ ShuffleNetV1 | resnet32x4 ↓ ShuffleNetV2 | WRN-40-2 ↓ ShuffleNetV1 |
| | | | training diverges | | | 52.81 | 57.99 | 53.31 |
| | | ✓ | 74.01 | 73.41 | 75.89 | 75.49 | 76.48 | 76.34 |
| | ✓ | | 74.42 | 73.48 | 75.97 | 75.80 | 76.45 | 75.96 |
| **IPWD** | ✓ | ✓ | **74.64** | **73.91** | **76.03** | **76.03** | **76.61** | **76.61** |

output. To evaluate the contribution of logits normalization, we considered an alternative that the logits are not normalized by the standard deviation. Results in Table 5 verify the contribution of each design. Without the classification head and logit normalization, the training is hard to converge or the performance is much worse. As a comparison, either the classification head or logit normalization helps with stable training. Besides, a combination of both further improves the performance and achieves the best results. The crash of training is due to the high variance of sample weights. Since the teacher model is well pre-trained and has more parameters, it has a larger capacity than the student model. Differently, an extra head with a shared backbone guarantees a similar capacity. Also, the normalization will avoid an extremely large or small CE loss, which further reduces the variance.

**Teacher trained with label smoothing.** Recent works [40, 51] observed that KD performs poorly with label smoothing. Similar to KD, the performance of IPWD drops when the teacher model is trained with label smoothing, but still outperforms KD. However, we found that the improvement of IPWD compared to KD also decreases with label smoothing. For example, on CIFAR-100, given ResNet50 as teacher and MobileNetV2 as student, IPWD outperforms KD by 1.12% (69.67% vs. 68.55%) without label smoothing, but the improvement drops to 0.56% (66.79% vs. 66.23%) with label smoothing. Given resnet32x4 as teacher and ShuffleNetV1 as student, IPWD outperforms KD by 1.52% (75.79% vs. 74.27%) without label smoothing, but the improvement drops to 0.53% (73.27% vs. 72.74%) with label smoothing. We observed that teacher trained with label smoothing produces more balanced predictions compared to teacher trained without label smoothing. Therefore, the results are consistent with our hypothesis that IPWD helps to bridge the transfer gap especially when the context information of teacher is imbalanced.

## 5.3 One-stage Self-Distillation

**Baseline methods and metrics**. For one-stage self-distillation, we apply our method to the state-of-the-art PS-KD [28] method as a plug-in, and consider label smoothing (LS) method and two self-distillation methods, CS-KD [68] and TF-KD [67], as baselines. PS-KD proposed a one-stage framework that progressively distills the knowledge of a model itself to soften the one-hot supervisions as regularization. The knowledge is transferred using a conventional distillation loss. As for metrics, besides top-1 and top-5 accuracy, we follow Kim *et al.* [28] to report expected calibration error (ECE, %) and the area under the risk-coverage curve (AURC, $\times 10^3$). A low ECE indicates well-calibrated predictions, and a low AURC represents the well-separation of correct and incorrect predictions.

**Implementation**. We follow all the training details of PS-KD for a fair comparison. Specifically, the architectures we considered are ResNet-18 [16], ResNet-101 [15], ResNeXt-29 [63] (cardinality=8, width=64), and DenseNet-121 [22] (growth rate=32). During training, PS-KD gradually determines how much the student learns from the teacher's knowledge. The formulation is:

$$\mathcal{L}_{ps\text{-}kd} = (1 - \alpha_t) \cdot \mathcal{L}_{cls} + \alpha_t \cdot \mathcal{L}_{dist}, \quad (8)$$

where the trade-off parameter $\alpha_t = \alpha_T \times t/T$, $T$ is the number of total epochs (*e.g.*, 300), $t$ is the current epoch, and $\alpha_T$ is a hyperparameter. Compared to Eq. (8), our IPWD applied on PS-KD is formulated as

$$\mathcal{L}_{ps\text{-}kd+ipw} = (1 - \alpha_t) \cdot \mathcal{L}_{cls} + \alpha_t \cdot \mathcal{L}_{ipw\text{-}dist} \quad (9)$$

Table 6: Results on CIFAR-100 test set for the one-stage self distillation setting over four architectures. Top-1 and Top-5 indicate the accuracy.

| Method | Top-1 | Top-5 | ECE | AURC | Method | Top-1 | Top-5 | ECE | AURC |
|---|---|---|---|---|---|---|---|---|---|
| ResNet-18 | 75.82 | 93.10 | 11.84 | 67.65 | DenseNet-121 | 79.95 | 95.01 | 7.34 | 52.21 |
| + LS | 79.06 | 93.98 | 10.79 | 57.74 | + LS | 80.20 | 94.54 | **0.92** | 91.06 |
| + CS-KD [68] | 78.70 | 94.30 | 6.24 | 56.56 | + CS-KD [68] | 79.53 | 93.79 | 13.80 | 73.37 |
| + TF-KD [67] | 77.12 | 93.99 | 11.96 | 61.77 | + TF-KD [67] | 80.12 | 94.90 | 7.33 | 69.23 |
| + PS-KD [28] | 79.18 | 94.90 | 1.77 | 52.10 | + PS-KD [28] | 81.27 | **96.10** | 3.71 | 45.55 |
| **+ PS-KD + Ours** | **79.82** | **95.15** | **1.39** | **49.71** | **+ PS-KD + Ours** | **81.60** | 96.04 | 3.48 | **45.33** |
| ResNet-101 | 79.25 | 94.72 | 10.02 | 55.45 | ResNeXt-29 | 81.35 | 95.53 | **4.17** | 44.27 |
| + LS | 80.16 | 94.93 | 3.43 | 95.76 | + LS | 82.40 | 95.77 | 22.14 | 41.92 |
| + CS-KD [68] | 79.24 | 94.38 | 12.18 | 64.44 | + CS-KD [68] | 81.74 | 95.63 | 5.95 | 42.11 |
| + TF-KD [67] | 79.90 | 94.90 | 6.14 | 58.80 | + TF-KD [67] | 82.67 | 96.13 | 6.73 | 40.34 |
| + PS-KD [28] | 80.57 | 95.70 | 6.92 | 49.01 | + PS-KD [28] | 82.72 | 96.40 | 9.15 | 39.78 |
| **+ PS-KD + Ours** | **81.39** | **95.91** | **3.19** | **43.82** | **+ PS-KD + Ours** | **83.30** | **96.60** | 4.93 | **37.49** |

Since both the student and teacher models are poor at early epochs, the weight estimation is not accurate at early epochs, which may lead to a worse self-teacher. Therefore, we apply IPWD at the last 75 epochs over the total 300 epochs.

**Comparison with baseline methods**. Table 6 shows the results of one-stage self-distillation methods over four architectures. Our IPWD can effectively and constantly improve the top-1 accuracy of PS-KD by 0.33%∼0.82% with different architectures. Besides, our IPWD significantly lowers the ECE and AURC of PS-KD. These results demonstrate the effectiveness of our IPWD.

**Ablation study: IPWD stage**. We conduct an ablation study to analyze whether IPWD should be started from an early stage (*e.g.*, the beginning of training) or a late stage (*e.g.*, last 1/4 of the epochs). We take ResNeXt-29 as an example. As shown in Table 7, applying IPWD from the beginning slightly outperforms PS-KD and under-performs the student modal that applies IPWD only at the late stage by large margins. As the student model is poorly trained at the early stage, the weight estimation is inaccurate and hurts the performance of self-teacher. These results indicate that the quality of estimated weight and distillation performance relies on the student model and self-teacher.

Table 7: Ablation study on the start of applying IPWD.

| Method | Top-1 Acc | Top-5 Acc | ECE | AURC |
|---|---|---|---|---|
| PS-KD [28] | 82.72 | 96.40 | 9.15 | 39.78 |
| + IPWD (early) | 82.86 | 96.35 | 8.56 | 38.16 |
| + IPWD (late) | **83.30** | **96.60** | **4.93** | **37.49** |

## 6   Conclusion

In this paper, we point out that conventional KD methods hold an invalid IID assumption and does not properly address the transfer gap between the context-invariant human domain and the context-equivariant machine domain, especially the imbalanced knowledge of the teacher model on the transfer set. We further proposed a simple yet effective method, Inverse Probability Weighting Distillation (IPWD), to deal with the imbalanced knowledge caused by transfer gap. In the future, we will extend our IPWD to (1) tasks beyond classification, like detection and segmentation, and (2) long-tailed tasks where the training samples in the human domain are also non-IID.

## Acknowledgment

We thank anonymous ACs and reviewers for their valuable discussion and insightful suggestions. This research is supported by the National Research Foundation, Singapore under its AI Singapore Programme (AISG Award No: AISG2-RP-2021-022) and Alibaba-NTU Singapore Joint Research Institute (JRI).

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
