# Appendix for "Respecting Transfer Gap in Knowledge Distillation"

**Yulei Niu**[1]    **Long Chen**[1]    **Chang Zhou**[2]    **Hanwang Zhang**[3]
[1]Columbia University   [2]Damo Academy, Alibaba Group   [3]Nanyang Technological University
{yn.yuleiniu,zjuchenlong}@gmail.com   zhouchang.zc@alibaba-inc.com
hanwangzhang@ntu.edu.sg

# Appendix

## 1  Experimental Settings

### 1.1  Two-stage Distillation

**Implementation**. We follow the training details of Tian *et al.* [18] for CIFAR-100 and Zhou *et al.* [23] for ImageNet. Specifically, for CIFAR-100, we set the mini-batch size as 64 and an initial learning rate as 0.05. We train the model for 240 epochs. The learning rate is decayed by 10 every 30 epochs after 150 epochs. We initialize the learning rate as 0.01 for MobileNetV2, ShuffleNetV1 and ShuffleNetV2, and as 0.05 for other models. The experiments are conducted using one NVIDIA TITAN RTX GPU. For ImageNet, we train the model for 100 epochs. We set the mini-batch size as 256, an initial learning rate as 0.1, and decay it by 10 every 30 epochs. The experiments are conducted using four Tesla V100 GPUs.

**Architectures**. We follow Tian *et al.* [18] for the choice of network architectures. Specifically, "WRN-d-w" denotes Wide Residual Network (WRN) [21] with depth $d$ and width factor $w$. "resnet-d" represents cifar-style ResNet [3] with 3 groups of basic blocks, each with 16, 32, and 64 channels respectively. For example, resnet8x4 and resnet32x4 represent a 4 times wider network, *i.e.*, with 64, 128, and 256 channels respectively. "ResNet-d" represents the ImageNet-style ResNet with Bottleneck blocks and more channels. MobileNetV2 [16] has a width multiplier of 0.5. "vgg" denotes VGGNet [17] that is adapted from its original ImageNet counterpart. "ShuffleNetV1" [22], "ShuffleNetV2" [10] are adapted with input size as 32x32.

### 1.2  One-stage Self-Distillation

**Implementation**. We follow all the training details of PS-KD [9]. The standard data argumentation schemes are 32x32 random crop after padding with 4 pixels and random horizontal flip. The networks are trained for 300 epochs using SGD with a momentum of 0.9. The learning rate is decayed by 10 at 150 and 225 epochs. We set the mini-batch size as 128, an initial learning rate as 0.1, and a weight decay as 0.0005. The experiments are conducted using four Tesla V100 GPUs.

**Metrics**. Besides top-1 and top-5 accuracies, we follow Kim *et al.* [9] and report Expected calibration error (ECE) and Area under risk-coverage curve (AURC) for evaluation. Expected calibration error (ECE) [11] is used to evaluate the confidence calibration performance of models, *i.e.*, the expected gap between accuracy and confidence. Specifically, the samples are partitioned by confidence into $M$ bins $B_1, \cdots, B_M$. The $m$-th bin $B_m$ contains samples with confidence within $\left[\frac{m-1}{M}, \frac{m}{M}\right]$. For $N$ samples, ECE is formulated as:

$$\text{BCE} = \frac{1}{N} \sum_{m=1}^{M} |B_m| \times |\text{Acc}(B_m) - \text{Conf}(B_m)|,$$

Table 1: Top-1 accuracy of student networks on CIFAR-100 test set for the two-stage distillation setting. * denotes our reproduced results using the same teacher model.

| | Same architecture style | | Different architecture style | |
|---|---|---|---|---|
| Teacher | WRN-40-2 | resnet110 | resnet32x4 | vgg13 |
| Student | WRN-16-2 | resnet20 | ShuffleNetV2 | MobileNetV2 |
| Teacher | 75.61 | 74.31 | 79.42 | 74.64 |
| Student | 73.26 | 69.06 | 71.82 | 64.60 |
| FitNet [15] | 73.58 | 68.99 | 73.54 | 64.14 |
| AT [20] | 74.08 | 70.22 | 72.73 | 59.40 |
| SP [19] | 73.83 | 70.04 | 74.56 | 66.30 |
| CC [14] | 73.56 | 69.48 | 71.29 | 64.86 |
| VID [1] | 74.11 | 70.16 | 73.40 | 65.56 |
| RKD [12] | 73.35 | 69.25 | 73.21 | 64.52 |
| PKT [13] | 74.54 | 70.25 | 74.69 | 67.13 |
| AB [4] | 72.50 | 69.53 | 74.31 | 66.06 |
| FT [8] | 73.25 | 70.22 | 72.50 | 61.78 |
| NST [6] | 73.68 | 69.53 | 74.68 | 58.16 |
| KD [5] | 74.92 | 70.67 | 74.45 | 67.37 |
| CRD [18] | 75.48 | **71.46** | 75.65 | 69.73 |
| WSLD* [23] | 75.63 | 71.20 | 75.55 | 68.50 |
| **IPWD** | **75.83** | 71.22 | **76.61** | **69.81** |

where $\mathrm{Acc}(B_m)$ denotes the accuracy of samples in $B_m$, and $\mathrm{Conf}(B_m)$ denotes the average confidence of samples in $B_m$. A lower BCE indicates a well-calibrated model.

Area under risk-coverage curve (AURC) [2] measures how well predictions are ordered by confidence values. Specifically, we can determine a threshold for classification, where only samples with confidence higher than the threshold are accepted. After that, we can obtain the proportion of covered samples to the whole dataset, *i.e.*, coverage, and define the risk as the error rate computed by using the covered samples. AURC is defined as the area under the risk-coverage curve. A lower AURC indicates that the correct and incorrect predictions are well-separable by confidence values.

## 1.3 License of Assets

We reimplemented WSLD[1] and SSKD[2] based on their open-resourced codes. Both WSLD and SSKD did not mention the license in their open-resourced codes.

## 2 Additional Results

### 2.1 Architecture styles

Due to page limitation, we did not include all the results of different architecture styles on CIFAR-100 in the main paper. We provide additional results in Table 1. Results show that our IPWD achieves competitive performances and outperforms KD by large margins.

### 2.2 Feature-based Methods

Note that our proposed IPWD is a logit-based distillation method. An interesting question is whether the reweighting strategy can work with feature-based distillation methods. We select ReviewKD as an example, which is a recent representative feature-based distillation method. As shown in Figure 2, the gap between ReviewKD+IPWD and ReviewKD is very marginal, which indicates that IPWD cannot promote feature-based distillation. The possible reasons are two-fold. First, the logit knowledge of label $y$ is long-tailed but the representation knowledge of sample $x$ may be relatively balanced. Second, as pointed out by Kang *et al.* [7], "data imbalance might not be an issue in learning

---

[1] https://github.com/bellymonster/Weighted-Soft-Label-Distillation
[2] https://github.com/xuguodong03/SSKD

Table 2: ReviewKD with our IPWD reweighting strategy on feature level.

| Teacher
Student | WRN-40-2
WRN-16-2 | resnet56
resnet20 | resnet110
resnet32 | resnet32x4
ShuffleV2 | WRN-40-2
ShuffleV1 |
|---|---|---|---|---|---|
| ReviewKD | 76.12 | **71.89** | **73.89** | **77.78** | **77.14** |
| ReviewKD + IPWD | **76.25** | 71.51 | 73.79 | 77.74 | 77.06 |

Table 3: Ablation study of CE-aware output on CIFAR-100. $^*$ denotes that the CE-aware output is obtained from an extra student model.

| Teacher
Student | resnet110
resnet32 | resnet32x4
resnet8x4 | resnet32x4
ShuffleV1 | resnet32x4
ShuffleV2 |
|---|---|---|---|---|
| Teacher | 74.31 | 79.42 | 79.42 | 79.42 |
| Student | 71.14 | 72.50 | 70.50 | 71.82 |
| IPWD$^*$ | 73.64 | 75.88 | 75.98 | 76.83 |
| **IPWD** | 73.91 | 76.03 | 76.03 | 76.61 |

high-quality representations" for long-tailed classification, which implies that the reweighting strategy is not compatible at feature level.

## 2.3 Long-tailed Methods for KD

Following LA, we applied the class prior to the student output when calculating the KL divergence distillation loss. We found that KD+LA underperforms KD by averagely 0.5% on CIFAR-100. The possible reason is that the introduced prior indirectly breaks the teacher's knowledge for each training sample, which hurts the effectiveness of distillation. These results indicate that logit-adjust-based long-tailed techniques are not applicable to the issue of KD.

## 2.4 Ablation Studies

We have conducted ablation studies on the components of our proposed IPWD. In the appendix, we further provide ablation studies on the technical designs.

We take two-stage distillation on CIFAR-100 as an example. Recall that the sample weights are estimated based on the two types of student's outputs, KD-trained output $y^{kd}$ from the student's original head and CLS-trained output $y^{cls}$ from the student's extra head. We further conduct ablations on the selection of two outputs.

For CE-aware output $y^{cls}$, an straightforward alternative is using an extra model that has the same architecture as student with totally different parameters. This extra model is trained using the cross-entropy classification loss. In other words, the difference is whether the visual backbone is shared for the two outputs. We denote this alternative as IPWD$^*$. Table 3 shows the comparison. Overall, IPWD$^*$ achieves competitive results compared to IPWD. However, training an extra model leads to more memory and time cost. Therefore, our design that takes an extra head is both effective and efficient.

For KD-trained output $y^{kd}$, an straightforward alternative is using an extra distillation head like the classification head. Different from the original head of the student, the distillation head is trained only using the distillation loss, which simply mimics the teacher's output without ground-truth annotations. We denote this alternative as IPWD$^\dagger$. Table 4 shows the comparison. Overall, IPWD$^\dagger$ slightly underperforms IPWD with different four architectures. The reason is that the extra distillation head only learns from the teacher model, which is sensitive to the teacher's performance and weights hard samples more. These two ablation studies further verify the effectiveness of our techinical designs.

Table 4: Ablation study of KD-aware output on CIFAR-100. [†] denotes that the KD-aware output is obtained from another extra head trained only with the original distillation loss.

| Teacher
Student | resnet110
resnet32 | resnet32x4
resnet8x4 | resnet32x4
ShuffleV1 | resnet32x4
ShuffleV2 |
|---|---|---|---|---|
| Teacher | 74.31 | 79.42 | 79.42 | 79.42 |
| Student | 71.14 | 72.50 | 70.50 | 71.82 |
| IPWD[†] | 73.56 | 75.88 | 75.93 | 76.44 |
| **IPWD** | 73.91 | 76.03 | 76.03 | 76.61 |