# OpenReview forum: "Respecting Transfer Gap in Knowledge Distillation"
_NeurIPS.cc/2022/Conference — NeurIPS 2022 Accept_

### Official Review · Reviewer_ADzo · 2022-07-11

**Rating:** 4
**Confidence:** 3
**Soundness:** 3 good
**Presentation:** 3 good
**Contribution:** 2 fair

**Summary:**

This paper proposes a new loss function for teacher-student learning by observing the fact that the label predictions from a teacher model has a distribution shift compared with ground-truth labels. It is pointed out that this distribution mismatch can cause a gap during transfer due to over-representation and under-representation of examples, e.g. the existence of context-equivariance and context-invariance information in some image datasets. Specifically, inverse probability weighting is used in the loss function to assign larger weights to the KL divergence loss of under-represented examples. The weight is implemented by comparing the cross-entropies of both classification outputs and knowledge distillation outputs to the ground-truth label. Finally, empirical evaluations are conducted on some image datasets and results show that the proposed method can achieve improved performance over some baselines.

**Questions:**

Q1: It is claimed several times in the paper that the dataset generated by the teacher model is no longer IID (e.g. L142). I don't understand why this is the case. Once we have a teacher model, the dataset is generated from the distribution induced by this teacher model. Given the parameter $\theta^t$, individual examples $(x,y^t)$ are also independent from each other. As introduced in Section 2, IPW is practically suitable to non-IID observations. The resulting distribution is likely to be long tailed but the examples still seem to be independent, which weakens the justification of the use of IPW. Could the authors please explain more on this?

Q2: L146: "Thanks to the causality-based theory, we can use the IPW technique to overcome the confounding effect brought by the transfer gap". In my opinion, this claim is not well supported by current writing. Which part of the causality-based theory is relevant to the problem described in this paper? In this paragraph only references are mentioned but detailed explanation is missing. For the second half of the sentence, I don't understand what is the confounding effect brought by the transfer gap. How does the confounding effect occur in the dataset of teacher model?

Q3: In Fig.1, the blue dashed line refers to the histogram of different labels. The ImageNet dataset (ILSVRC 2012) has 1,000 classes with varying number of images per class ranging from around 732 to 1300. Its training set is imbalanced. I don't see why in the right subfigure the label distribution is a straight line. How is the average probability on the y-axis calculated for labels or predictions from a teacher model?

Q4: In experimental comparisons, the proposed method is compared with some baselines including CRD and WSLD. I'm not sure whether the reported results are SOTA on knowledge distillation benchmark. Are there any other published baselines?


**Limitations:**

See weakness and questions.

**Strengths And Weaknesses:**

Strengths: The motivation of solving the distribution mismatch problem of the teacher model is reasonable and novel. This observation is of practical importance in knowledge distillation as this problem is expected to occur in many teacher student learning datasets. The use of inverse probability weighting (IPW) for KD in the proposed method seems to be helpful for this problem according to the results of experiments and ablation study.

Weaknesses: Although the aforementioned problem is well clarified, the use of IPW is not well motivated or explained. Many explanations of IPW and its motivation relate to the problem of non-IID in the model dataset. However, my concern is that this does not exactly fit or target the problem of distribution mismatch problem. Please see the questions below for more details. Moreover, the method of IPW has been well-established before. The technical contribution is limited.

---

> ### Author Response · Authors · 2022-08-02
> **Response to Reviewer ADzo (Part 1/2)**
>
> **Q1: Given the parameter $\theta^t$, are $(x, y^t)$ independent from each other? If so, the justification of the use of IPW is weakened.**
>
> A: No, $(x,y^t)$ are non-independent from each other. Note that hard targets $y$ only provides context-invariant class-specific information (Line 39), while $y^t$ further contains context information. As the teacher model is trained on the training set $\mathcal{D}$, the parameter $\theta^t$ is conditioned on all the training samples $(x,y)$. As $y^t$ is obtained by $y^t=f(x;\theta^t)$, $y^t$ is conditioned on $\theta^t$, and further conditioned on other training pairs $\mathcal{D}\setminus${$(x,y)$} (Line 47-49). In other words, $y^t$ is obtained by seeing other training samples. Therefore, $(x, y^t)$ are non-independent from each other, and samples with frequent context information are more likely to be observed in the transfer set.
>
> ---
>
> **Q2: What is the confounding effect brought by the transfer gap and occur in the dataset of teacher model?**
>
> A: We are sorry for the confusion. We provided the detailed explanations of confounder, confounding effect, and the effect of IPW in the following.
>
> The training data $\mathcal{D}=${$(x,y)$} and teacher model $\theta^t$ jointly act as the confounder of image $x$ and teacher prediction $y^t$ in the transfer set. First, the training set $\mathcal{D}$ and transfer set of teacher model $\mathcal{D}^t=${$(x,y^t)$} share the same image set, and $x$ is sampled from the image set of $\mathcal{D}$, i.e., $\mathcal{D}$ serves the cause of $x$. Second, the teacher $\theta^t$ is trained on $\mathcal{D}$, and $y^t$ is calculated based on $\theta^t$ and $x$, i.e., $y^t=f(x;\theta^t)$. Therefore, $x$ and $\theta^t$ are the cause of $y^t$. Note that the transfer set is constructed based on the images on $\mathcal{D}$ and teacher model $\theta^t$. Therefore, we regard $\mathcal{D}^t$, the joint of $\mathcal{D}$ and $\theta^t$, as the confounder of $x$ and $y^t$.
>
> Although $\mathcal{D}$ is balanced when considering the context-invariant class-specific information, the context information (e.g., attributes) is overlooked, which makes the $\mathcal{D}$ imbalanced in context. Such imbalanced context leads to an imbalanced transfer set $\mathcal{D}^t$ (as shown in Figure 1 in the main paper), and further affects the distillation of teacher's knowledge.
>
> To overcome the above confounding effect, a commonly used technique is intervention via $P(y^t|do(x))$ instead of $P(y^t|x)$, which is formulated as $P(y^t|do(x))=\sum_{\mathcal{D}^t} P(y^t|x,\mathcal{D}^t)P(\mathcal{D}^t)=\sum_{\mathcal{D}^t} \frac{P(x,y^t,\mathcal{D}^t)}{P(x|\mathcal{D}^t)}$. This transformation suggests that we can use the inverse of propensity score $P(x|\mathcal{D}^t)$ (i.e., $P(x|machine)$ in the main paper) as sample weight to implement the intervention and overcome the confounding effect.
>
> ---
>
> **Q3: Why is the label distribution of ImageNet a straight line? How is the average probability on the y-axis calculated for labels or predictions from a teacher model?**
>
> A: We agree and have noticed that the training set of ImageNet is not perfectly balanced. However, 895 out of 1000 classes have 1300 images, and only 36 classes have less than 1100 images. Therefore, we think the training set is still relatively balanced. For simplicity, we use a straight line to denote the relative balance of training set and highlight the imbalance of teacher predictions. We will update the figure to avoid misleading.
>
> For calculating the average probability from a teacher model, we first obtain the teacher's predicted probability distribution for each training sample, and then sum up the probabilities over all the training samples. The simplification of training distribution on ImageNet does not affect the correctness and sharpness of teachers' prediction distribution.

---

> > ### Comment · Reviewer_ADzo · 2022-08-09
> > **Thank you for response**
> >
> > I would like to thank the authors for the detailed response to my earlier questions. The response clarifies my concerns over independence and confounding effect. Overall, I think the technical contribution and empirical performance (e.g. on ImageNet) are not so significant. I would like to increase my rating by 1.

---

> > > ### Author Response · Authors · 2022-08-09
> > > **Further responses to technical contribution and empirical performance**
> > >
> > > Thank you for the acknowledgement of our responses and for upgrading the rating! For your remaining concerns, we would like to summarize the motivation, contributions (especially the **technical contribution** of propensity score estimation), and empirical performance on ImageNet in the following:
> > >
> > > * **Motivation**: We proposed to review KD in the perspective of transfer gap. We found that (1) the teacher's knowledge is imbalanced on the transfer set (Figure 1 and Line 51-61 in the revised version), and (2) the distillation performance of the under-represented classes is the bottleneck (Response to Reviewer 6yEk's W1). We hope to compensate for the under-weighted training samples.
> > >
> > > * **Analytical contributions**: We proposed IPWD inspired by the success of inverse probability weighting (IPW) in causal inference. We **interpret the transfer gap in KD from the perspective of causal inference**, and pointed out that the transfer set brings in the confounding effect (Line 148-169 in the revised paper).
> > >
> > > > The method of IPW has been well-established before. The technical contribution is limited.
> > >
> > > * **Technical contributions**: Although IPW is widely used in causal inference, how to apply IPW to KD remains challenging but has not been exploited. The key challenges are: (1) how to implement the idea of intervention and IPW, and (2) how to estimate the propensity score as there are no annotations. To tackle the first challenge, we **proposed a weighted distillation loss** (Eq. (6)) to compensate for the under-weighted training samples and overcome the confounding effect. To tackle the second challenge, we **proposed a new propensity score estimation strategy** to obtain the sample weights from data automatically. Specifically and technically, we (1) added an extra classification-trained (CLS-trained) head (Figure 3(b)) and compared the outputs of the CLS-trained head and KD-trained head (Eq. (5)), and (2) normalized the logits for stable training. We believe that our implementation of propensity score estimation (Line 202, Line 209, Eq. (5) in the revised paper) is novel and detailed, and the ablation studies in Table 5 validated the effect of our technical designs.
> > >
> > > > Empirical performance (e.g. on ImageNet) is not so significant.
> > >
> > > * **Performance on ImageNet**: We believe that the results on ImageNet validate the motivation, hypothesis, and effectiveness of our IPWD well. The results on ImageNet are three parts. First, the results with the **same architecture style for two-stage distillation**. Our IPWD **slightly underperforms WSLD by 0.16\%** (71.88\% vs. 72.04\%) **but outperforms other baselines**, including the recent logit-based distillation method DKD published in CVPR'22. Second, the results with the **different architecture style for two-stage distillation**. Our IPWD **outperforms WSLD by 1.03\%** (72.65\% vs. 71.52\%), **and outperforms recent DKD by 0.6\%** (72.65\% vs. 72.05\%). We believe **the results with different architecture styles**, i.e., the performance gap between teacher and student is larger (3.56\% vs. 7.29\%), **are more convincing to validate our motivation and hypothesis of transfer gap**. Similar results can be observed on CIFAR-100. Third, the results with the **for one-stage distillation**. Our IPWD **consistently promote PS-KD on various metrics** (Table 6). In addition to increasing the accuracies, our IPWD significantly lowers the ECE (\%) and AURC (×10$^3$) metrics (lower is better), which are used to indicate the calibration and separation of predictions. For example, for ResNet-101, IPWD lowers the ECE of PS-KD from 6.92 to 3.19, and lowers the AURC from 49.01 to 43.82.
> > >
> > > ---
> > >
> > > Thank you again for your effort and time in improving our work! We hope the further responses can address the your remaining concerns. Please let us know if there is anything unclear.

---

> ### Author Response · Authors · 2022-08-02
> **Response to Reviewer ADzo (Part 2/2)**
>
> **Q4: Any other published baselines?**
>
> A: For fair comparisons, we compare our method with baseline methods using the same teacher model parameters, whether the teacher model may be the bottleneck of the student model's performance. One of the state-of-the-art methods SSKD achieves high performance with a better pre-trained teacher (e.g., improving teacher model WRN-40-2 from 75.61 to 76.46). In the main paper, we have also reimplemented SSKD and applied our IPWD on top of SSKD using the same teacher parameters. As shown in Table 2 in the main paper, our IPWD can improve SSKD with various architecture styles.
>
> Another recent logit-based state-of-the-art method DKD (CVPR'22) is officially published after the NeurIPS submission deadline.
> We reimplemented DKD (denoted as DKD$^*$) and IPWD using their released code and the sample teacher model for fair comparisons.
>
> As shown in the below table, our IPWD achieves comparable performances on CIFAR-100, and outperforms DKD on ImageNet. In particular, IPWD outperforms DKD by 0.6\% with ResNet-50 as teacher and MobileNet-v1 as student on ImageNet, i.e., teacher and student have different architecture styles. These comparisons demonstrate the effectiveness of our IPWD.
>
> *Table: Results on CIFAR-100*
>
> | Teacher  |   resnet50 |  resnet32x4  |	resnet32x4	|  WRN-40-2	|  vgg13 |
> |-------------------|----------------|----------------|----------------|----------------|----------------|
> | Student  |   MobileNetV2 |   ShuffleNetV1    |	ShuffleNetV2	|    ShuffleNetV1	|  MobileNetV2 |
> |   DKD |   **70.35**   |	76.45 |	**77.07** |	76.70  |	69.71 |
> |   DKD$^*$ |   70.27	| 76.03   |	76.99 |	76.49 |	69.02 |
> |   IPWD    |   69.78	| **76.61**   |	76.72 |	**76.92** |	**69.81** |
>
> *Table: Results on ImageNet*
>
> | Teacher  |   ResNet-34  |  |  ResNet-50  |   |
> |----|------|----------|--------|-------|
> | Student  |   ResNet-18  |  |  MobileNet-v1 |   |
> |   |   Top-1  | Top-5 |  Top-1 | Top-5 |
> |   DKD |   71.70   |	90.41 |   72.05 |	91.05  |
> |   IPWD    |   **71.88**	| **90.50**   |	**72.65** |   **91.08** |
>
> [SSKD] Knowledge Distillation Meets Self-Supervision. ECCV'20.
>
> [DKD] Decoupled Knowledge Distillation. CVPR'22.

---

> ### Author Response · Authors · 2022-08-07
> **Welcome to discuss**
>
> Dear Reviewer ADzo,
>
> We sincerely thank you for your efforts and time in our work. We tried our best to address all the concerns and questions. We have also updated the main paper and appendix following your comments. Please feel free to let us know if you have any further concerns or questions to discuss.
>
> Best,
>
> Paper 2438 Authors

---

> > ### Author Response · Authors · 2022-08-09
> > **Kind reminder of author-reviewer discussion deadline (August 9)**
> >
> > Dear Reviewer ADzo,
> >
> > Just a kind reminder that the author-reviewer discussion will end soon on Tuesday, August 9. Reviewer 6yEk has acknowledged our rebuttal and raised the rating. If you have any further questions or concerns, please don't hesitate to let us know! We are looking forward to your follow-up feedback.
> >
> > Best,
> >
> > Paper 2438Authors

---

### Official Review · Reviewer_6yEk · 2022-07-12

**Rating:** 5
**Confidence:** 4
**Soundness:** 2 fair
**Presentation:** 3 good
**Contribution:** 3 good

**Summary:**

This paper proposes a new KD method named IPWD. The authors state that the teacher's knowledge is imbalanced due to the imbalanced soft labels in teacher's predictions, and propose to use inverse probability weighting (IPW) to balance the weight of each sample in KD loss. Experiments on CIFAR-100 and ImageNet are provided to show the superiority of IPWD.

**Questions:**

1. The distribution of teacher's soft labels would be varied a lot according to different training strategies. Can IPWD adapt better to these distribution shifts? For example, some works [1, 2] observed that KD performs poorly when with label smoothing.

I will raise my rating if the authors can well address my concerns.


[1] Müller, R., Kornblith, S. and Hinton, G.E., 2019. **When does label smoothing help?**. Advances in neural information processing systems, 32.
[2] Shen, Z., Liu, Z., Xu, D., Chen, Z., Cheng, K.T. and Savvides, M., 2020, September. **Is Label Smoothing Truly Incompatible with Knowledge Distillation: An Empirical Study**. In International Conference on Learning Representations.

**Limitations:**

Both limitations and potential negative societal impact were discussed.

**Strengths And Weaknesses:**

Strengths:
1. This paper gives an interesting domain transfer perspective of KD, in which the distributions of ground-truth labels and teacher's soft labels are different, this conflict makes KD less effective. Based on this, the authors propose a simple KD loss to balance the teacher knowledge using IPW, which gains significant improvements over the KD baseline.

Weaknesses:
1. The authors should discuss the impact of imbalanced teacher knowledge in conventional KD methods, to show the necessity of balanced knowledge. For example, compare the predictions of under-represented class of student trained with or without KD.
2. In Table 3, IPWD performs worse than WSLD on ResNet-18 student (71.88% vs. 72.04%).

---

> ### Author Response · Authors · 2022-08-02
> **Response to Reviewer 6yEk**
>
> **W1: What is the impact of imbalanced teacher knowledge in conventional KD methods, e.g., comparing the predictions of under-represented class of student trained with or without KD?**
>
> A: Thanks for pointing this out. Following the reviewer's suggestion, we rank and divide the 100 classes of CIFAR-100 into 4 groups according to the averaged predicted probability of the teacher model on the training set. Following the long-tail recognition task that also groups classes according to their numbers of samples, we take the macro-average recall as the metric. We report the improvement of KD compared to vanilla training (i.e., student trained without KD) in the below table.
>
> Compared to vanilla training, KD achieves better performance in all the subgroups. However, going deeper into the improvement for each subgroup, we found that the increase in the top 25 classes (i.e., over-represented) is much higher than the last 25 classes (i.e., under-represented), i.e., averagely 5.14 vs. 0.85. This observation verified our hypothesis that the effectiveness of KD on the under-represented samples is the bottleneck of KD, which is an interesting but overlooked issue in existing works.
>
> | Teacher->Student           |   Top 1-25            | Top 26-50             | Top 51-75              | Top 76-100              |
> |-------------------|----------------|----------------|----------------|----------------|
> | ResNet50 -> MobileNetV2             | +4.96          | +5.92         | +1.76          | +1.20          |
> | resnet32x4 -> ShuffleNetV1             | +5.80          | +2.68        | +2.52           | +0.84          |
> | resnet32x4 -> ShuffleNetV2             | +4.72          | +1.92         | +2.24          | +0.76          |
> | WRN-40-2 -> ShuffleNetV1             | +5.08          | +7.20         | +4.48          | +0.60          |
>
> ---
>
> **W2: In Table 3, IPWD performs worse than WSLD on ResNet-18 student (71.88\% vs. 72.04\%) on ImageNet.**
>
> A: Actually, our IPWD performs better than WSLD in most cases on CIFAR-100 and ImageNet.
>
> On CIFAR-100, IPWD outperforms WSLD with various teacher and student architectures, especially when their architecture styles are different, i.e., the gap between teacher and student is large. On ImageNet, although our IPWD underperforms WSLD by 0.16\% when the teacher and student have the same architecture style (ResNet-34 -> ResNet-18) and their performance gap is relatively small (top1 accuracy gap: 3.56\%), IPWD outperforms WSLD by 1.13\% when their architecture style are different (ResNet-50 -> MobileNet-v1) and their performance gap is relatively large (top1 accuracy gap: 7.09\%).
>
> These comparisons demonstrate the effectiveness of IPWD to bridge the transfer gap, especially when the gap between teacher and student models is large, We believe this setting is more practical and general in real-world applications. (Lines 253-267)
>
> ---
>
> **Q1: Can IPWD work with distribution shifts of teacher models caused by label smoothing?**
>
> A: Similar to KD, the performance of IPWD drops when the teacher model is trained with label smoothing, but still outperforms KD. However, we found that the improvement of IPWD compared to KD also decreases with label smoothing.
>
> For example, on CIFAR-100, given ResNet50 as teacher and MobileNetV2 as student, IPWD outperforms KD by 1.12\% (69.67\% vs. 68.55\%) without label smoothing, but the improvements drop to 0.56\% (66.79\% vs. 66.23\%) with label smoothing. Given resnet32x4 as teacher and ShuffleNetV1 as student, IPWD outperforms KD by 1.52\% (75.79\% vs. 74.27\%) without label smoothing, but the improvement drops to 0.53\% (73.27\% vs. 72.74\%) with label smoothing.
>
> We observed that teacher trained with label smoothing produces more balanced predictions compared to teacher trained without label smoothing. Therefore, the results are consistent with our hypothesis and conclusion that IPWD helps to bridge the transfer gap especially when the context information of teacher is imbalanced.

---

> > ### Comment · Reviewer_6yEk · 2022-08-09
> > **Response to Authors**
> >
> > Thanks for your detailed responses.
> >
> > **W1**: The results is convincing to me. I hope the authors can add this into the revised paper to better clarify your motivation.
> >
> > **W2**: Maybe a combination of IPWD and recent state-of-the-art KD methods can achieve better accuracy, it is nice to see that in the future revision.
> >
> > **Q1**: I suggest the authors to provide experiments on ImageNet in the future revision. Also, Reviewer T8CL raises a interesting topic about stronger teachers, the authors could have more experiments and discussions on it to show the superiority of IPWD.

---

> > > ### Author Response · Authors · 2022-08-09
> > > **Thank you for the rebuttal acknowledgement**
> > >
> > > Thank you for the follow-up comments and suggestions! We have include the discussion about W1 in the revised paper. We will follow your suggestions and are working on the application to more SOTA KD methods, more results on ImageNet, and discussion with strong teachers. We will keep updating once we have new results.

---

> ### Author Response · Authors · 2022-08-07
> **Welcome to discuss**
>
> Dear Reviewer 6yEk,
>
> We sincerely thank you for your efforts and time in our work. We tried our best to address all the concerns and questions. We have also updated the main paper and appendix following your comments. Please feel free to let us know if you have any further concerns or questions to discuss.
>
> Best,
> Paper 2438 Authors

---

### Official Review · Reviewer_T8CL · 2022-07-12

**Rating:** 5
**Confidence:** 5
**Soundness:** 3 good
**Presentation:** 3 good
**Contribution:** 3 good

**Summary:**

This paper investigates the knowledge distillation problem by formulating it as a transfer learning problem. It finds that even if the teacher is trained with balanced training data, the transferred information could be imbalanced and it hurts the performance of KD. Therefore, this paper proposes Inverse Probability Weighting Distillation (IPWD) as an improvement of typical knowledge distillation loss, and extensive experimental results show the effectiveness of the proposed IPWD.

**Questions:**

Please refer to the "Weaknesses".

**Limitations:**

Yes.

**Strengths And Weaknesses:**

+ The paper investigates KD from a transfer learning perspective, which is interesting compared to the typical dark knowledge.
+ The structure and writing are good.
+ The experimental results are comprehensive.

- Figure 1 shows that there seems long-tailed property of teacher predictions. I wonder whether it could be fixed by some techniques from long-tailed classification.

- Formulation of propensity in Section 4.2 seems sort of heuristic. I do not really get how to understand it in a principal and mathematical way.

- IPDW is a logits-based KD method. Would it be better if combined with the feature-based method? Necessary discussions and experimental comparisons are needed.

- This paper claims that knowledge distillation will work if there is a valid transfer gap. However, recent work argues that if the teacher becomes way too strong, the KD seems to degrade. Will the proposed IPDW handle this issue? If so, experiments about a strong teacher should be included. A recent work DIST investigates a similar setting, which should be helpful if the authors want to verify the transfer gap issue in terms of a huge or strong teacher.

Knowledge Distillation from A Stronger Teacher, https://arxiv.org/abs/2205.10536

---

> ### Author Response · Authors · 2022-08-02
> **Response to Reviewer T8CL**
>
> **W1: Can techniques for long-tailed classification fix the long-tailed property of teacher predictions?**
>
> A: Thanks for the insightful question! We select LA (ICLR'21) as the recent representative technique for long-tailed classification. LA proposed a logit adjusted softmax cross-entropy loss by applying a class prior to each logit. LA does not require extra modules (compared to TDE, NeurIPS'20), post-hoc logit adjustment (compared to LADE, CVPR'21), or ensemble of multiple models (compared to RIDE, ICLR'21).
>
> Following LA, we applied the class prior to the student output when calculating the KL divergence distillation loss. We found that KD+LA underperforms KD by averagely 0.5\% on CIFAR-100. The possible reason is that the introduced prior indirectly breaks the teacher's knowledge for each training sample, which hurts the effectiveness of distillation. These results indicate that logit-adjust-based long-tailed techniques are not applicable to the issue of KD. We will explore other types of long-tailed techniques in the future.
>
> [LA] Long-tail learning via logit adjustment. ICLR'21.
>
> [TDE] Long-Tailed Classification by Keeping the Good and Removing the Bad Momentum Causal Effect. NeurIPS'20.
>
> [LADE] Disentangling Label Distribution for Long-tailed Visual Recognition. CVPR'21.
>
> [RIDE] Long-Tailed Recognition by Routing Diverse Distribution-Aware Experts. ICLR'21.
>
> ---
>
> **W2: How to understand propensity in a principal and mathematical way?**
>
> A: Let us first introduce the concepts of confounder in distilling teacher's knowledge. The training data $\mathcal{D}=${$(x,y)$} and teacher model $\theta^t$ jointly act as the confounder of image $x$ and teacher prediction $y^t$ in the transfer set. First, the training set $\mathcal{D}$ and transfer set of teacher model $\mathcal{D}^t=${$(x,y^t)$} share the same image set, and $x$ is sampled from the image set of $\mathcal{D}$, i.e., $\mathcal{D}$ serves the cause of $x$. Second, the teacher $\theta^t$ is trained on $\mathcal{D}$, and $y^t$ is calculated based on $\theta^t$ and $x$, i.e., $y^t=f(x;\theta^t)$. Therefore, $x$ and $\theta^t$ are the cause of $y^t$. Note that the transfer set is constructed based on the images on $\mathcal{D}$ and teacher model $\theta^t$. Therefore, we regard $\mathcal{D}^t$, the joint of $\mathcal{D}$ and $\theta^t$, as the confounder of $x$ and $y^t$.
>
> Although $\mathcal{D}$ is balanced when considering the context-invariant class-specific information, the context information (e.g., attributes) is overlooked, which makes the $\mathcal{D}$ imbalanced in context. Such imbalanced context leads to an imbalanced transfer set $\mathcal{D}^t$ (as shown in Figure 1 in the main paper), and further affects the distillation performance of teacher's knowledge.
>
> To overcome such confounding effect, a commonly used technique is intervention via $P(y^t|do(x))$ instead of $P(y^t|x)$, which is formulated as $P(y^t|do(x))=\sum_{\mathcal{D}^t} P(y^t|x,\mathcal{D}^t)P(\mathcal{D}^t)=\sum_{\mathcal{D}^t} \frac{P(x,y^t,\mathcal{D}^t)}{P(x|\mathcal{D}^t)}$. This transformation suggests that we can use the inverse of propensity score $P(x|\mathcal{D}^t)$ (i.e., $P(x|machine)$ in the main paper) as sample weight to implement the intervention and overcome the confounding effect.
>
> ---
>
> **W3: Can IPWD work with feature-based distillation method?**
>
> A: Perhaps not. ReviewKD is a recent representative feature-based distillation method. We applied IPWD on ReviewKD at the feature level. We found that IPWD slightly decreases the performance of ReviewKD in most cases with marginal gaps, which indicates that IPWD is not applicable to feature-based distillation.
>
> | Teacher  |   WRN-40-2 |  resnet56  | resnet110 |  resnet32x4	|  WRN-40-2 |
> |--------|------|----|-----|-----|-----|
> | Student  |   WRN-16-2 |   resnet20    |	resnet32	|    ShuffleNetV2	|  ShuffleNetV1 |
> | ReviewKD |   76.12   |	**71.89** | **73.89** |	**77.78**  |	**77.14** |
> | ReviewKD+IPWD |   **76.25**	| 71.51   |	73.79 |	77.74 |	77.06 |
>
> The possible reasons are two-fold. First, although the logit knowledge of label $y$ is imbalanced, the representation knowledge of sample $x$ might be relatively balanced. Second, as pointed out by Decouple for long-tailed classification, "data imbalance might not be an issue in learning high-quality representations", which implies that the reweighting strategy is not compatible at the feature level.
>
> We will include this discussion in the revised version and add it to limitation discussion.
>
> [ReviewKD] Distilling Knowledge via Knowledge Review. CVPR'21.
>
> [Decouple] Decoupling Representation and Classifier for Long-Tailed Recognition. ICLR'20.
>
> ---
>
> **W4: Can IPWD work with a huge or strong teacher?**
>
> A: Thanks for the interesting research question. Note that the reference is an ArXiv preprint released after the NeurIPS submission deadline. We are happy to include this work in our future work discussion.

---

> > ### Comment · Reviewer_T8CL · 2022-08-09
> > **Some of my concerns are resolved.**
> >
> > I really appreciate the response from the authors. However, I am concerned with your response to W3. First, the authors claim that "although the logit knowledge of label is imbalanced, the representation knowledge of sample might be relatively balanced". Does this mean that we simply use feature-based KD instead of logits-KD? Besides, I am not convinced by the results of combined feature-based KD method. I wonder whether the weight for the feature-based KD term is well set.

---

> > > ### Author Response · Authors · 2022-08-09
> > > **Follow-up responses**
> > >
> > > Thank you for the further comments!
> > >
> > > > Does this mean that we simply use feature-based KD instead of logits-KD?
> > >
> > > We did not mean that. Logits-based strategies and feature-based strategies are two different research directions for KD. Our main focus is how to promote **logit-based distillation** methods (e.g., KD in Tables 2 and 4, SSKD in Table 3, PS-KD in Table 6 in our main paper) from the perspective of transfer gap, especially the imbalanced teacher's **logit knowledge**. As our analyses are all at the logit level, there is no guarantee that our analyses and conclusions on the logit level can be directly adopted to the feature level, and we won't over-claim the generalization of our conclusions.
> > >
> > >  > I wonder whether the weight for the feature-based KD term is well set.
> > >
> > > We would like to provide more details of implementing IPW on ReviewKD. The weights for feature-based KD term are calculated in the same way as logit-based KD, i.e., Line 212 and Eq. (6) in our revised paper. Note that the weight calculation in our IPWD are based on the outputs of a KD-trained classifier (Line 197) and a CLS-trained classifier (classification-trained classifier, Line 196, 204-205). Since the classifier of ReviewKD is updated using cross-entropy loss, we added an extra classifier which is trained with KD loss, and the gradients of KD loss would not be back-propagated to the visual backbone. The calculated sample weights are appended to the feature-based KD term for each sample. During the test stage, the extra classifier is discarded.
> > >
> > > A more narrowed and strict conclusion is that the weight calculation in our IPWD is not applicable to feature-based methods. We think the reweighting strategy for feature-based KD would be an interesting but less exploited direction, which is out of the scope of our paper. Inspired by the discussion with the reviewer, we would like to explore other reweighting strategies for feature-based KD in the future.
> > >
> > > -------
> > >
> > > Please let us know if your have any further questions, concerns, or suggestions!

---

> ### Author Response · Authors · 2022-08-07
> **Welcome to discuss**
>
> Dear Reviewer T8CL,
>
> We sincerely thank you for your efforts and time in our work. We tried our best to address all the concerns and questions. We have also updated the main paper and appendix following your comments. Please feel free to let us know if you have any further concerns or questions to discuss.
>
> Best,
>
> Paper 2438 Authors

---

### Author Response · Authors · 2022-08-02
**Response to all reviewers**

First of all, we gratefully thank all the reviewers for their thoughtful comments and feedback.

We are encouraged that the reviewers find our proposed domain transfer perspective of KD interesting (Reviewer T8CL, Reviewer 6yEk), reasonable and novel (Reviewer ADzo). We are glad that the reviewers find our paper has good structure and writing (Reviewer T8CL), our proposed method is helpful (Reviewer ADzo) and achieves gains significant improvements over the KD baseline (Reviewer 6yEk), and our experimental results are comprehensive (Reviewer T8CL).

We tried to address all the concerns and questions in detail. In particular, following the suggestions and comments of reviewers, we further provide (1) detailed explanations of propensity and confounding effect (Reviewer T8CL, Reviewer ADzo), (2) a discussion about applying long-tailed techniques on KD (Reviewer 6yEk), (3) a discussion about applying our method on feature-distillation (Reviewer 6yEk), (4) an analysis of KD's performance on under-represented classes (Reviewer 6yEk), and (5) the effect of training the teacher model with label smoothing on our method (Reviewer 6yEk).

Hope that our response answers the questions.

---

### Meta-Review · Area_Chair_tyPr · 2022-08-26

**Recommendation:** Accept
**Confidence:** Less certain

**Metareview:**

This paper analyzes the way in which most previous knowledge distillation methods violate IID assumptions and it aims to address the drop in performance on student models through this analysis. The paper proposes an Inverse Probability Weighting Distillation (IPWD) technique, derived in part through a causal analysis of the distillation setting. Results are mainly presented for CIFAR-100, but some ImageNet results are given and these result show that the proposed approach does indeed outperform a wide variety of prior work for distillation. The review scores for this paper place it right at the borderline of acceptance, with two weak accepts and one weak reject.

Given the paper was at the borderline of numerical acceptance and the signals from reviews and subsequent discussions were not conclusive, the Area Chair also read this paper and found the underlying idea to be quite interesting and novel. The application of causal analysis to the problem in this way does a nice job of brining together an important branch of machine learning (causal analysis) with deep learning and knowledge distillation. The AC also judged that the experimental work in this paper was substantial. Given that the method also yields better results than many other prior methods, AC recommends accepting this paper.


**Award:**

No

---

### Decision · Program_Chairs · 2022-09-14

Accept